# Learning CT Segmentation from Label Masks Only

**Artyom Tsanda**[1,2]                                    ARTYOM.TSANDA@TUHH.DE
**Hannes Nickisch**[3]                                   HANNES.NICKISCH@PHILIPS.COM
**Tobias Wissel**[3]                                      TOBIAS.WISSEL@PHILIPS.COM
**Tobias Klinder**[3]                                     TOBIAS.KLINDER@PHILIPS.COM
**Tobias Knopp**[1,2,4]                                   TOBIAS.KNOPP@TUHH.DE
**Michael Grass**[3]                                      MICHAEL.GRASS@PHILIPS.COM

[1] *Hamburg University of Technology, Institute for Biomedical Imaging, Germany*

[2] *University Medical Center Hamburg-Eppendorf, Section for Biomedical Imaging, Germany*

[3] *Philips Innovative Technologies, Germany*

[4] *Fraunhofer Research Institution for Individualized and Cell-Based Medical Engineering IMTE, Germany*

## Abstract

Training segmentation models for CT scans in the absence of input data is a challenging problem. Methods based on generative adversarial networks translate images from other modalities but still require additional data and training. Synthesizing images directly from segmentation masks using heuristics can overcome this limitation. However, capabilities for model generalization remain underexplored for these methods. In this study, we generate synthetic data for liver segmentation using organ labels and prior CT knowledge. Ground truth labels serve as a source of information about global structures and are filled with artificial textures in various settings. Segmentation models trained on synthetic data demonstrate sufficient generalization to real CT data, highlighting a perspective of a simple yet powerful approach to data bootstrapping.

**Keywords:** Semantic Segmentation, CT, Synthetic Data Generation

## 1. Introduction

Paired datasets of CT images and ground truth (GT) segmentations are not always available prohibiting training of deep neural networks (DNN). To address the issue of data scarcity, labeled data from other imaging modalities can be used. For instance, using cycle generative adversarial network CT values can be estimated from MRI data (Zhang et al., 2018; Huo et al., 2019). Alternatively, a dataset can be extended using mask-to-image translation approaches based on conditional generative adversarial models (Gheorghiță et al., 2022). The label-driven approach allows to apply augmentations to labels and obtain a paired input. Both methods rely on an additional generative network for image synthesis, which in turn requires data and can be cumbersome to train.

Synthesizing images directly from segmentation masks using heuristics (imperative approaches) can be sufficient for a DNN to generalize to the target domain. Billot et al. (2020, 2023) employed a Gaussian mixture model and extensive augmentations for image synthesis. Varying parameters of the Gaussian distribution for each tissue, the authors generated a multi-contrast and cross-modal dataset for brain and cardiac segmentation. The trained segmentation models could generalize to variable MRI contrasts, CT and PET data. Based on this method, Hoopes et al. (2022) proposed a segmentation model for skull-stripping.

Hoffmann et al. (2022) successfully trained a DNN for contrast-invariant MRI registration using randomly generated and post-processed structures within brain masks.

In this paper, we introduce several imperative methods for generating synthetic CT data for semantic segmentation derived from labels. We consider the problem of liver segmentation as a benchmark for the proposed methods. The results indicate that segmentation models trained on the synthetic data generalize to real CT scans, demonstrating performance comparable with that of the model trained on real data. This study highlights a new perspective for data generation in the anatomical segmentation of CT images.

## 2. Materials and methods

Our study is based on the TotalSegmentator dataset (Wasserthal et al., 2023) for segmentation of human organs on CT images. We consider only abdomen CT scans with full-body coverage and large FOV acquired in full-dose setting which results in 64 training, 17 validation and 44 test cases (see Appendix A). Using segmentation labels and prior CT-specific knowledge, we generate three synthetic training and validation datasets with different number of included labels and texture variability.

**Synthetic data #1: single label.** A cylinder resembling the human body with a random radius is filled with a histogram-matched video from the Inter4K (Stergiou and Poppe, 2021) dataset providing diverse backgrounds. The video is reflection-padded along the time dimension to fit the target volume. Histogram matching is performed with the body region of a reference CT scan. Voxels from the liver mask are filled with a random constant value drawn from the Gaussian distribution and pasted into the generated cylinder. The distribution parameters are estimated using CT data from the train set. Unlike Billot et al. (2020, 2023), we do not randomize the distribution parameters as the problem of variable contrast is less pronounced in CT images and the values can be well described using a single distribution.

**Synthetic data #2: labels filled with random constant values.** This option requires each voxel to be segmented, and since the TotalSegmentator leaves a few structures unsegmented, the rest of the CT volume is first filtered with a threshold to segment air and then clustered into two classes using the K-means algorithm. A similar approach was proposed in Billot et al. (2020, 2023) where the clustering was used not only to complement the given labels but also to subdivide the foreground into multiple subclasses. We assign constant values to voxels from the resulting 3 classes and fill back voxels segmented by the TotalSegmentator with a random value assigned to each organ, similar to the previous approach. The resulting CT volume represents a piecewise constant CT image.

**Synthetic data #3: labels filled with natural videos.** In the last setting, we follow the same steps from the second dataset but fill the labels with histogram-matched videos, providing both rich structural and versatile textural information.

As an augmentation, for all three datasets with a 50% chance a label is not pasted back to the volume. For each case from the training set, we generate 4 synthetic scans. Finally, we simulate CT-specific noise by forward-projecting volumes onto helical geometry, converting the projections into photon counts, and sampling new photon counts from the Poisson distribution. After the noise simulation, the projection data are reconstructed using the Aperture Weighted Wedge Filtered Backprojection algorithm (Koken and Grass, 2006).

The results of data generation for all three methods are shown in Fig. 1 where for the last method (Fig. 1(c)) the liver is not pasted back, corresponding to a synthetic patient without the liver. Although for this study we used a dataset of real CT scans, the proposed strategies remain applicable if labels come from another image modality. Histograms and distribution parameters for each organ can be considered as prior knowledge.

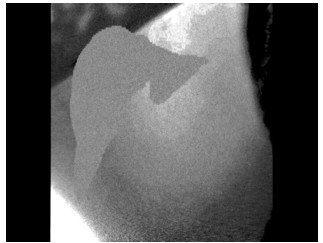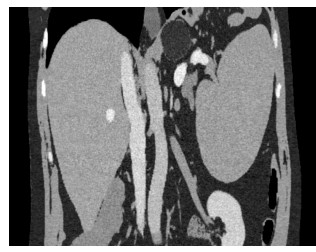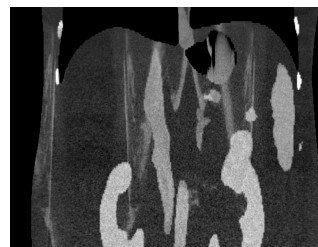

(a) Single label.    (b) Labels filled with random constant values.    (c) Labels filled with natural videos.

Figure 1: Examples of synthetically generated CT images.

For each dataset, we trained a segmentation model using the nnU-Net framework (Isensee et al., 2021). Pre-processing parameters were re-used from the original dataset, *i.e.*, normalization parameters. The model was set up for high image resolution in 3D and trained using a single split into training and validation for 4000 epochs.

The resulting models are assessed on real CT scans from the test set. We filter the largest connected component as a post-processing step for the resulting segmentations. The quality of segmentation results is estimated using averaged Dice scores for the target organ.

## 3. Results and Conclusion

Table 1 shows averaged Dice scores for models trained on the considered synthetic datasets and a model trained on real CT scans from the training set. As the models demonstrate comparable performance, the ones trained on synthetic data, with less information employed, successfully generalize to real CT data. Moreover, the model trained using only information about the shape of the liver (Synthetic data #1) is on par with the others, meaning higher importance of the shape than the surrounding in the case of liver segmentation.

Table 1: Dice scores

| Dataset | Clinical data | Synthetic data #1 | Synthetic data #2 | Synthetic data #3 |
|---------|---------------|-------------------|-------------------|-------------------|
| **Dice score** | 0.947 | 0.951 | 0.971 | 0.967 |

In this study, we proposed several strategies to generate synthetic CT segmentation datasets that rely on segmentation labels and prior knowledge about CT images. We trained segmentation models on synthetic data to segment the liver and showed that the models can generalize to real CT scans. The results indicate a promising approach to bootstrap CT segmentation data as it does not require actual images and employs an imperative strategy without the need to train proxy-models. Future work should assess the segmentation performance of other organs and potentially lesions.

## Acknowledgments

This project is funded by the Deutsche Forschungsgemeinschaft (DFG, German Research Foundation) – SFB 1615 – 503850735

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

## Appendix A. Cases selected from the TotalSegmentator dataset

In this work, we consider the following cases from the TotalSegmentator dataset:

- Training cases:
  s0472, s0091, s0796, s0763, s0649, s0507, s0467, s1208, s0456, s1006, s0196, s1369, s1230, s0644, s0662, s1099, s1247, s0617, s0916, s0896, s0913, s0790, s0628, s1404, s1336, s0835, s1319, s0970, s0028, s0950, s0447, s1070, s1145, s0578, s1224, s0429, s1314, s0945, s0358, s0334, s0612, s0899, s0591, s0836, s0484, s0076, s1120, s0362, s0878, s1082, s0939, s1085, s1061, s0553, s0549, s1159, s0519, s0863, s1012, s1350, s0065, s0765, s1063, s0903.

- Validation cases:
  s0961, s0669, s0476, s0369, s0461, s0992, s1111, s1044, s1210, s1348, s0797, s1031, s0592, s1089, s0957, s0764, s0983.

- Test cases:
  s0013, s0029, s0038, s0040, s0045, s0119, s0235, s0236, s0244, s0291, s0308, s0311, s0423, s0440, s0441, s0450, s0458, s0499, s0505, s0543, s0546, s0561, s0607, s0625, s0650, s0667, s0687, s0794, s0918, s0923, s0933, s0991, s0994, s1046, s1152, s1161, s1174, s1223, s1233, s1238, s1249, s1276, s1287, s1377.

