# OpenReview forum: "Learning CT Segmentation from Label Masks Only"
_MIDL.io/2024/Short_Papers — MIDL 2024 Short Papers_

### Official Review · Reviewer_V5gD · 2024-04-16

**Confidence:** 4
**Final Rating:** 4

**Review:**

Summary:
The authors present three strategies to generate synthetic training data based on segmentation maps and prior knowledge about CT. The synthetic images are obtained by construction from the segmentations by either using 1) liver labels only with backgrounds obtained from videos 2) full segmentations where each label is filled with a random intensity 3) full segmentation maps where labels are filled with the content from a video. These three models are able to succesfully generalise to real CTs.

Weaknesses:
- Missing references: The authors cite [1] and [2], but even more relevant papers are [3,4] (which in fact inspired [1,2]), where synthetic images are created from label maps to train a segmentation network, with applications to MRI and CT brain and cardiac segmentation. These papers must be cited.
- Filling segmentations with random values like for synthetic data #1 and #2 directly comes from [3], which must be given credit. Also, complementing the available segmentations with background labels obtained with clustering has already been proposed by [4], which must be cited in Synthetic data #2.
- Missing baseline: Can the authors comment why they fill the labels with constant values rather than sampling from random label-specific Gaussians like in [1,2,3,4] ? because sampling from Gaussians enable the simulation of scanner noise. Also, the generative model of [1,2,3,4] should have been tested as a baseline.
- Clarifications: The construction of the first synthetic dataset is obscure and needs better explanations.

[1] Hoffmann et al., SynthMorph: Learning Contrast-Invariant Registration Without Acquired Images. IEEE TMI, 2022

[2] Hoopes et al., SynthStrip: skull-stripping for any brain image. NeuroImage, 2022

[3] Billot et al., A Learning Strategy for Contrast-agnostic MRI Segmentation, MIDL, 2020

[4] Billot et al., SynthSeg: Segmentation of brain MRI scans of any contrast and resolution without retraining”. Medical Image Analysis, 2023.

---

### Decision · Program_Chairs · 2024-04-26

Accept